# SimReg: Achieving Higher Convergence and Generalization in the LLM Pretraining via Embedding Similarity Regularization

## Abstract

Pretraining large language models (LLMs) with next-token prediction has led to remarkable advances, yet the context-dependent nature of token embeddings in such models results in high intra-class variance and inter-class similarity, thus hindering the efficiency of representation learning. While similarity-based regularization has demonstrated benefit in supervised fine-tuning and classification tasks, its application and efficacy in large-scale LLM pretraining remains underexplored. In this work, we propose the SimReg, an embedding similarity regularization loss that explicitly encourages token representations with the same ground-truth label within each sequence to be more similar, while enforcing separation from different-label tokens via a contrastive loss. Our comprehensive theoretical analysis elucidates how SimReg improves both classification margins and generalization in the pretraining stage. Extensive experiments across dense and Mixture-of-Experts (MoE) architectures demonstrate that SimReg consistently accelerates training convergence by over $30\%$ and improves average zero-shot downstream performance by over $1\%$ across standard benchmarks. Further ablation and analysis provide practical recommendations for hyperparameter selection and loss application, offering constructive insights for efficient pretraining of LLMs.

## 1 Introduction

LLMs have emerged as a cornerstone of modern artificial intelligence and have demonstrated remarkable capabilities across a wide range of domains such as natural language understanding (Radford et al., 2019), reasoning (Wei et al., 2022), and multimodal interaction (Lin et al., 2025). While LLMs are advancing along diverse directions, they all fundamentally share a consistent underlying principle, i.e., next-token prediction. The essential mechanism of LLMs is to predict the categorical distribution of the next token from the embeddings of the prior context, which can also be viewed as a classification problem defined over the combined representations of the preceding context. By leveraging enormous model parameters and vast training data, it exhibits exceptional generalization capability, introduces novel solutions across diverse research domains, and further drives the adoption of a wide range of applications (Topsakal & Akinci, 2023).

However, in contrast to conventional classification problems, this prediction paradigm does not rely on a stable and intrinsic object that strictly match its label. For example, in image classification tasks, the image of a cat is bound to its label during training, and the embeddings of the same class exhibit very high consistency. While in the autoregressive training paradigm, an object of the predicted token can be regarded as a composition of other entities that are irrelevant to the current label, which leads to a practical challenge: the token embeddings used to predict the same label can differ substantially. As an illustrative example, in the following two samples, "The cat jumps over walls" and "A child paints near walls", the predictive embeddings for "walls" come from entirely different sources, considerably complicating the pretraining process.

Recent advances in consistency learning for finetuning language models shed light on potential solutions to this challenge. Leveraging label-aware constructive losses to regularize specific network features for sufficient discriminability has been shown to enhance generalization on domain-specific tasks (Huang et al., 2021; Gunel et al., 2021; Yin et al., 2023). However, this line of research has

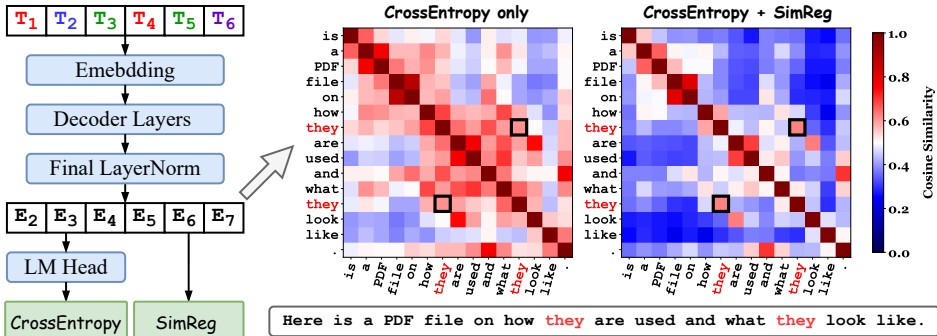

Figure 1: (left) Workflow of the SIMREG loss. (Right) We compare the cosine similarity of token embeddings in a sample on the LLaMA-7B model trained via "CrossEntropy only" and "CrossEntropy+SIMREG". It can be clearly observed that using cross-entropy only fails to effectively enforce sufficient separability among features of different labels, whose cosine values of all token pairs exceed 0.5. With the introduction of SIMREG, feature separability is generally enhanced (averaged cosine value is reduced by at least 0.1), thereby providing stronger support for classification training. We also provide more visualization demos of training samples from C4 dataset in Appendix B.4.

not yet been extended to pretraining and has not been widely adopted in the large-scale pretraining practices. Post-training is typically performed with a small learning rate and limited datasets, which makes it hard to change the overall nature of the model. These insights motivate us to further extend this approach to large-scale pretraining, where it has the potential to yield general benefits.

only In this work, we show that large-scale pretraining with cross-entropy alone fails to impose strong consistency on token embeddings. To address this, we then add a consistency regularization term, SIMREG, to strengthen the representational capacity of large models during pretraining. For each token in a sequence, all tokens are partitioned into positive and negative groups. The objective penalizes the similarity across groups, which pulls embeddings toward same-class samples and pushes them away from different-class samples. To ensure valid contrastive pairs for every token, SIMREG introduces self-sample similarity in each positive group and further computes the loss with group-level rather than sample-level averaging, which balances the contributions of different tokens, which allows it to preserve a high level of stability over the long pretraining runs. We also provide a thorough theoretical understanding to explain how it contributes to improving cross-entropy loss. Extensive evaluations are conducted on both dense and MoE models, including LLaMA-350M, 1.3B, 3B, 7B (Touvron et al., 2023), and Mixtral-8×1B (Jiang et al., 2024). The SIMREG loss can consistently accelerates convergence by over 30% in pretraining. When training with over 52B tokens, it can yield an improvement of more than 1% in average performance across downstream general tasks. We investigate the hyperparameter sensitivity of SIMREG and find that it maintains a wide range of applicability and elaborately explore its optimal insertion position in practice.

We summarize the core contributions of this work as follows:

- We explore the advantages of employing consistency regularization in large-scale pretraining tasks and propose a series of improvements to address the training instabilities of existing methods, thereby enabling stable performance gains throughout long-term pretraining.

- We provide a detailed theoretical analysis of the benefits of the SIMREG loss for the cross-entropy loss, and how it improves the multi-classification margins in the pretraining.

- We conduct extensive experiments to validate its substantial improvements for pretraining tasks, achieving an average training acceleration of over 30% and yielding over 1% gains on downstream tasks, and state detailed empirical insights for the community.

## 2 RELATED WORK

**Contrastive learning**. The systematic exploration of feature similarity constraints in machine learning can be traced back to their early development in computer vision (CV) tasks and contrastive

learning (Oord et al., 2018; Khosla et al., 2020). They enhance the training of baseline classification models by constructing virtual data pairs and incorporating additional supervised loss signals, which helped the models extract more discriminative features. It is typically employed to counteract noise perturbations at the input level, thereby improving generalization ability (Geng et al., 2021; Shi et al., 2022; Huang & Gong, 2022; Zhou et al., 2024; Wang et al., 2024). Generally, a data pair is constructed from a raw sample and its perturbed counterpart, and the model is trained to minimize their representation similarity. Subsequently, supervised contrastive learning has been extended to incorporate class information. By leveraging available labels to construct class-consistent data pairs, the model is trained not only to pull together samples from the same class but also to push apart samples from different classes (Wang & Liu, 2021; Wen & Li, 2021; Ye et al., 2022; Denize et al., 2023). Recent studies have revealed that contrastive learning can also achieve more efficient feature extraction across tasks and data originating from different domains (Verma et al., 2021; Wang et al., 2022; Xie et al., 2022; Azuma et al., 2023). In multimodal large model training, this learning paradigm is often employed to align the mapping of knowledge across domains and to capture the representation capacity of the same knowledge under different modalities (Yuan et al., 2021; Mai et al., 2022; Liu et al., 2024b; Sun et al., 2024). In summary, contrastive learning offers an efficient and general paradigm for representation learning to the machine learning community.

**Embedding Consistency in LLMs.** In LLM tasks, the study of feature similarity has also been considered as learning compositional generalization (Lake, 2019; Wiedemer et al., 2023) and embedding consistency regularization (Yin et al., 2023). Gao et al. (2021) learn the sentence embeddings and achieve higher generalization efficiency. Then it is widely expanded to the token-level (Gao et al., 2023; Wang & Yu, 2023), word-level (Kenter & De Rijke, 2015; Antoniak & Mimno, 2018), context-level (Laskar et al., 2020). Most of these tasks have primarily focused on small-scale training and fine-tuning settings. As the cornerstone of modern language models, the next-token prediction paradigm has been widely applied across various downstream tasks (Li et al., 2024; Chen et al., 2024). Recent research has further investigated the similarity and dispersion of token embeddings. In line with classification tasks in machine learning, the separability of embeddings has naturally become a key direction of study (de Andrade et al., 2023; Tao et al., 2024; Hu et al., 2024). By introducing label-aware data pairs to explicitly enforce the embedding properties across different groups, language models can gain additional benefits on both convergence and generalization.

## 3 PROBLEM SETUP

In this section, we introduce how consistency regularization can be incorporated into the pretraining of LLMs and explain why it helps improve both optimization efficiency and generalization performance. Before proceeding, we first formalize the overall pretraining setup of LLMs and introduce the notations used throughout the subsequent analysis.

**General Pretraining.** Before introducing the training framework, we first define the notation in this work. We consider the progress of LLM pretraining as learning the optimal weight $\mathbf{w}$ by minimizing the cross-entropy loss $\ell$ under a general distribution $\mathcal{D}$. We decompose the model into two cascaded functions $f_P \circ f_E$, where $f_P$ (the logits generation module) is parameterized by $\mathbf{w}_P$ and $f_E$ (the embedding generation module) is parameterized by $\mathbf{w}_E$, with the overall parameters denoted as $\mathbf{w} = [\mathbf{w}_P, \mathbf{w}_E]$. Based on this decomposition, the general pretraining objective of language models can then be formally formulated as the follows:

$$\min_{\mathbf{w}} L_{\text{ce}} \triangleq \mathbb{E}_{(\mathbf{x}_i, y_i) \sim \mathcal{D}} \left[ \ell \left( f_P \circ f_E \left( \mathbf{x}_i \right), y_i \right) \right], \tag{1}$$

where $(\mathbf{x}_i, y_i)$ is the (data, label) pair sampled from the distribution $\mathcal{D}$. Here, the choice of $f_E$ and $f_P$ is entirely flexible, meaning that the SIMREG loss can in principle be applied to any valid token embedding across the network. Nevertheless, our experiments indicate that attaching it directly to the output of the final layer yields the most consistent and significant performance improvements, highlighting the practical advantage of this placement.

**Limitations of Cross-Entropy in Language Models.** Cross-entropy loss serves as the fundamental training objective in language modeling. It measures the discrepancy between the predicted token distribution and the ground-truth one-hot distribution, thereby guiding the model to maximize the likelihood of the correct next token. The model typically employ large-scale feature extractors to obtain separable representations. By denoting the token embedding as $\mathbf{e}_i = f_E(\mathbf{x}_i)$ and corresponding

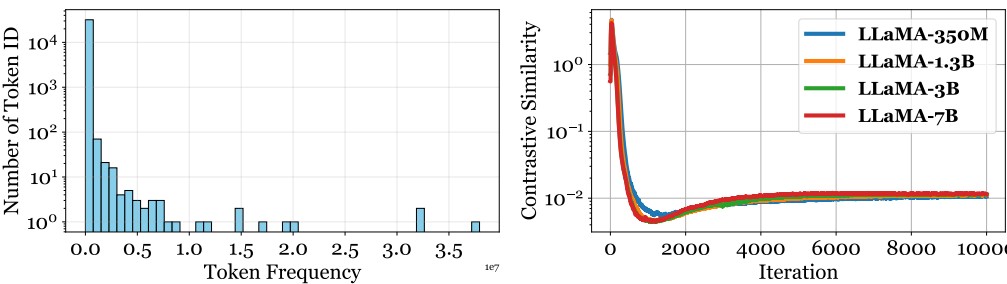

(a) Imbalanced Token Frequency in Pretraining.    (b) Limited Contrastive Similarity in Pretraining.

Figure 2: (a) We analyze the label of token ID distribution of 1B training samples from the C4 dataset and observe that only about 2% of the tokens occur with extremely high frequency, resulting in a pronounced long-tail effect in the classification data. (b) We observe that the contrastive similarity of embeddings at the pre-logits layer does not continue to decrease as the cross-entropy loss diminishes; instead, after reaching a certain threshold, the similarity is no longer further trained.

logits as $\mathbf{z}_i = f_P(\mathbf{e}_i)$, the population risk of sample-wise cross-entropy loss is:

$$L = \frac{1}{n} \sum_i \ell_i = \frac{1}{n} \sum_i \left( -\mathbf{z}_{i,y_i} + \log \left( \sum_j \exp \left( \mathbf{z}_{i,j} \right) \right) \right). \tag{2}$$

Generally, larger separability can enhance the distinction between different samples, leading to more robust and discriminative representations. Although Eq. (2) averages over samples, the unique characteristics of language tasks introduce a challenge: *the distribution of words (tokens) is highly imbalanced*, which causes frequent tokens to dominate the loss while rare but informative ones contribute disproportionately little, yielding a heavy long-tail dataset. When training classification tasks on such dataset, the inter-class margin is greatly influenced by the number of samples per class. As shown in Figure 2 (a), we empirical investigate the token distribution of C4 dataset and the behavior of contrastive similarity. **A primary challenge we investigate in LLM pretraining is that:**

Cross-entropy **stops** driving stronger representation learning after reaching basic separability.

Figure 2.(b) indicates that during the early stage of cross-entropy training, the model rapidly constrains the contrastive similarity of embeddings. However, once the contrastive diversity becomes sufficient to sustain classification training, the model no longer enforces heterogeneity among token embeddings. Subsequently, even though the cross-entropy loss continues to decrease, the contrastive similarity exhibits little further change. Another interesting phenomenon we observe is that, even as the model depth increases and the embedding dimension grows, the supervision of token embedding contrastive similarity under cross-entropy remains nearly at the same level. This limits the potential for further improvement in classification tasks, while also motivating us to impose stronger supervision on the contrastive similarity during LLM pretraining.

**Embedding Similarity Regularization.** Previous work has already explored this direction in small-scale training and fine-tuning tasks (Gao et al., 2021; Li et al., 2024). Here, we begin by defining a generalized form of the similarity regularization. The embedding can be denoted by $\mathbf{e} = f_E(\mathbf{x})$. For any two embeddings $\mathbf{e}_i$ and $\mathbf{e}_j$, we define a monotonic function $\phi(\mathbf{e}_i, \mathbf{e}_j)$. For each data pair $(\mathbf{x}_i, y_i)$, we can define a positive embedding set $\mathcal{P}_i = \{k : y_k = y_i\}$ and a negative embedding set $\mathcal{N}_i = \{k : y_k \neq y_i\}$. The core objective of the consistency loss is to pull the embeddings of positive pairs as close as possible. To maintain the numerical stability, we adopt the logexpsum formation:

$$\min_{\mathbf{w}_E} L_{\mathrm{sr}} \triangleq \mathbb{E}_{(\mathbf{x}_i, y_i) \sim \mathcal{D}} \left[ \log \left( \sum_{k \in \mathcal{N}_i} \phi(\mathbf{e}_i, \mathbf{e}_k) \right) - \log \left( \sum_{k \in \mathcal{P}_i} \phi(\mathbf{e}_i, \mathbf{e}_k) \right) \right]. \tag{3}$$

**Training with SimReg Loss.** For the similarity function $\phi$, we explore two primary forms: the exponential of the inner-product $\langle \mathbf{e}_i, \mathbf{e}_j \rangle$ and the cosine similarity $\frac{\langle \mathbf{e}_i, \mathbf{e}_j \rangle}{\|\mathbf{e}_i\| \cdot \|\mathbf{e}_j\|}$. Both similarity measures provide effective supervision for feature similarity, yet their applicable scenarios differ. It often

yields stronger statistical constraints, thereby enforcing supervision on both geometric structure and feature norms. However, this advantage may also introduce ambiguity: for instance, when a embedding has an abnormally large norm, the inner-product value becomes dominated by the magnitude, rendering the loss function almost insensitive to angular differences. In such cases, the optimization may overly rely on vector norms while neglecting the discriminative power of directional alignment. Therefore, for numerical stability, we adopt cosine similarity as the similarity measure in Eq. (3) and introduce a constant temperature coefficient $\tau$ to adjust the sharpness of the distribution. Since words in natural language are inherently distributed in an imbalanced manner, a sequence may contain only a single occurrence of a particular token type, we add the self-similarity to $\mathcal{P}_i$ to enforce that there exists at least a positive data pair. Moreover, to ensure that the regularization loss is non-negative, we introduce the softplus function to further scale it. Therefore, the final form of the loss is computed as $L = L_{\mathrm{ce}} + \lambda \cdot \mathrm{softplus}\,(L_{\mathrm{sr}})$. The entire optimization process involves two constant hyperparameters $\tau$ and $\lambda$. We discuss their optimal selections in Sec. 5.2.

**Chunk-wise SIMREG for Sequence Parallelism.** The computation of SIMREG is centered on the embedding of each token within a sequence sample, constructed in relation to all other tokens in the same sequence. Therefore, its complexity is $\mathcal{O}(n^2)$, where $n$ is the sequence length. Its computation can naturally support parallelization strategies like data parallelism (DP), tensor parallelism (TP), and pipeline parallelism (PP). However, during long-text training, sequence parallelism (SP) splits the data of each sequence across different nodes for training, which introduces additional redundant communication when the loss needs to be computed across nodes. To alleviate this issue, we directly divide SIMREG into $b$ chunks for computation, where every $\frac{n}{b}$ tokens form a chunk to compute the SIMREG loss internally. The losses across different nodes are then weighted according to the ratio of positive and negative samples, while the overall computational complexity is reduced to $\mathcal{O}(n^2/b)$. **Empirical results show that keeping** $1024$ **tokens per chunk is sufficient to maintain efficiency.**

## 4    THEORETICAL ANALYSIS

In this section, we present the theoretical analysis of the SIMREG loss. In particular, we establish its equivalent objective and demonstrate how this objective facilitates faster optimization for the cross-entropy loss. All proofs are provided in Appendix C.

### 4.1    HOW DOES THE SIMREG LOSS BENEFIT THE CROSS-ENTROPY LOSS?

We first analyze the embeddings of arbitrary independent samples after projection into the logits space. Specifically, we posit that their separability is not arbitrary but is inherently constrained, and can be bounded by the corresponding multi-classification margin, i.e., $m_k = \mathbf{z}_{k,y_k} - \max_{j \neq y_k} \mathbf{z}_{k,j}$. Then we can upper bound the sample-wise cross entropy loss by:

$$\ell_k = \log\left(1 + \sum_j \exp\left(\mathbf{z}_{k,j} - \mathbf{z}_{k,y_i}\right)\right) \leq \log\left(1 + \sum_j \exp\left(-m_k\right)\right) \leq C \exp\left(-m_k\right), \quad (4)$$

where $C$ is the number of total classes. The above formula explicitly characterizes the relationship between the classification margin and the training loss, from which it can be observed that enlarging the margin $m$ leads to a further reduction in the loss. This naturally motivates us to investigate:

- How does the SIMREG loss complement and benefit the cross-entropy objective?
- How much can the SIMREG loss accelerate the optimization performance?

**SIMREG enlarges classification margins.** To understand the performance of SIMREG in detail, we first introduce a general kernal function $\kappa\,(\mathbf{u}, \mathbf{v}) = \exp\left(\mathbf{u}^\top \mathbf{v}\right)$, which admits the Maclaurin series $\kappa\,(\mathbf{u}, \mathbf{v}) = \sum_{m=0}^{\infty} \frac{\left(\mathbf{u}^\top \mathbf{v}\right)^m}{m!}$. It is a positive definite kernel on the unit sphere. By introducing an explicit map: $h : \mathbb{S}^{d-1} \to \mathcal{H}$ on the symmetric tensor powers:

$$h(\mathbf{u}) = \left[1, \frac{1}{\sqrt{\pi}}\mathbf{u}, \frac{1}{\sqrt{2!\pi^2}}\mathrm{vec}\left(\mathbf{u}^{\otimes 2}\right), \frac{1}{\sqrt{3!\pi^3}}\mathrm{vec}\left(\mathbf{u}^{\otimes 3}\right), \cdots\right], \quad (5)$$

thus we have the transformation of $\langle h(\mathbf{u}), h(\mathbf{v})\rangle = \kappa\,(\mathbf{u}, \mathbf{v})$. The mapping $h$ is to construct a linear expansion of $\kappa$ in the reproducing kernel Hilbert space (RKHS) $\mathcal{H}$. Thus we have the follow lemmas.

**Lemma 4.1 (Objective)** *For each token $\mathbf{x}_i$ where its embedding is $\mathbf{e}_i = f_E(\mathbf{x}_i)$, according to the definition of positive and negative set in Eq. (3), we further define the corresponding kernel means by $\mu_k^+ = \frac{1}{|\mathcal{P}_k|} \sum_{i \in \mathcal{P}_k} h(\mathbf{e}_i)$ and $\mu_k^- = \frac{1}{|\mathcal{N}_k|} \sum_{j \in \mathcal{N}_k} h(\mathbf{e}_j)$ and thus objective in Eq. (3) equals to:*

$$\min_{\mathbf{w}_E} J \triangleq \mathbb{E}_{(\mathbf{x}_k, y_k) \sim \mathcal{D}} \left[ \log \left( \frac{\langle h(\mathbf{e}_k), \mu_k^- \rangle}{\langle h(\mathbf{e}_k), \mu_k^+ \rangle} \right) \right]. \tag{6}$$

Based on the above lemma, we can derive that the representation constraint of the SIMREG loss in the $\mathcal{H}$ space encourages each token embedding to move closer to the kernel mean of its homogeneous set while pushing it away from the kernel means of heterogeneous set. This mechanism effectively enhances intra-class compactness and inter-class separability in the embedding space. As a result, it facilitates more discriminative representations, which are crucial for improving classification robustness and generalization. Next, we illustrate the eventual impact of Eq. (6) on cross-entropy.

Our supervision on the embedding variable $\mathbf{e}$ is propagated through a function $f_P(\cdot)$ to the logits $\mathbf{z}$ used for the classification with cross-entropy, i.e., $\mathbf{z} = f_P(\mathbf{e})$. This mapping can take the form of a simple linear projection (e.g., the LM head) or several intermediate layers of the LLM. Without loss of generality, we assume it to be a general smooth and non-convex function with smoothness coefficient $L_P$. Thus, we consider the margin. By defining $\mathbb{I}$ as the standard basis vector where $\mathbb{I}_j$ means 1 in the $j$-th coordinate and 0 elsewhere, we measure the pair-wise gap in logits by $g_{y_k,j}(\mathbf{e}_k) = (\mathbb{I}_{y_i} - \mathbb{I}_j)^\top f_P(k)$, which also holds smoothness and non-convexity. Furthermore, we can transfer the smoothness by: $|g_{y_i,j}(\mathbf{e}_p) - g_{y_i,j}(\mathbf{e}_q)| \leq \|\mathbb{I}_{y_i} - \mathbb{I}_j\| \|f_P(\mathbf{e}_p) - f_P(\mathbf{e}_q)\| \leq \sqrt{2} L_P \|\mathbf{e}_p - \mathbf{e}_q\|$. To investigate the relationship between the margin and the embedding, we have the following lemma.

**Lemma 4.2 (Central–Eccentric Bound.)** *For each token $\mathbf{x}_i$ where its embedding is $\mathbf{e}_i = f_E(\mathbf{x}_i)$, we further define a weighted center of the embedding in the original space, where the positive and negative centers are $\overline{\mathbf{e}}_k^+ = \frac{\sum_{i \in \mathcal{P}_k} \alpha_{k,i} \mathbf{e}_i}{\sum_{i \in \mathcal{P}_k} \alpha_{k,i}}$ and $\overline{\mathbf{e}}_k^- = \frac{\sum_{i \in \mathcal{N}_k} \alpha_{k,i} \mathbf{e}_i}{\sum_{i \in \mathcal{N}_k} \alpha_{k,i}}$ where $\alpha_{k,i} \propto \kappa(\mathbf{e}_k, \mathbf{e}_i)$. Then we have the averaged group margins are $\overline{m}_k^+ = \min_{c \neq y_k} g_{y_k,c}(\overline{\mathbf{e}}_k^+)$ and $\overline{m}_k^- = \min_{c \neq y_k} g_{y_k,c}(\overline{\mathbf{e}}_k^-)$. Therefore, we can obtain that the classification margin bound of each token $m_k$ is the Central–Eccentric lower bound ithin the group margin:*

$$\overline{m}_k^+ - \sqrt{2} L_P \|\mathbf{e}_k - \overline{\mathbf{e}}_k^+\| \leq m_k \leq \overline{m}_k^- + \sqrt{2} L_P \|\mathbf{e}_k - \overline{\mathbf{e}}_k^-\|. \tag{7}$$

Intuitively, $\overline{m}$ can be regarded as an idealized margin, obtained by evaluating the logit of the correct class at the positive center and that of the strongest competing class at the negative center. Compared with evaluating directly at the embedding $\mathbf{e}$, this formulation is more effective because the centers are smoothed from many neighboring samples with exponential cosine weighting and this makes the margin a clearer and more stable measure of intra-class alignment and inter-class separation.

A more important point is that the above lemma separates the upper and lower bounds of the classification margin for each individual sample, showing that the lower bound is influenced by the distance to positive samples $\|\mathbf{e}_k - \overline{\mathbf{e}}_k^+\|$ and $\overline{m}_k^+$, while the upper bound is determined by the distance to negative samples $\|\mathbf{e}_k - \overline{\mathbf{e}}_k^-\|$ and $\overline{m}_k^-$. By minimizing Eq. (6), we can easily verify that:

- The central distance of positive set would decrease: $\frac{d}{dt} \|\mathbf{e}_k - \overline{\mathbf{e}}_k^+\|^2 \leq 0$;
- The margin at the positive center would increase: there exists a positive constant $\delta$ that

$$g_{y_k,j}(\mathbf{e}_k^+ + \epsilon_+) - g_{y_k,j}(\mathbf{e}_k^+) \geq \delta \|\epsilon_+\|, \tag{8}$$

where $\epsilon_+$ is the update with the gradient of the objective $J$ in Eq. (6).

By minimizing the objective $J$, we push the positive class mean $\mu_k^+$ away from the negative class mean $\mu_k^-$ in the RKHS, which induces two deterministic effects in input space: first, the kernel-weighted positive center $\overline{\mathbf{e}}_k^+$ shifts its weights toward same-class samples that are more similar to the anchor $\mathbf{e}_k$, causing $\|\mathbf{e}_k - \overline{\mathbf{e}}_k^+\|$ to decrease $\gamma$. Then the norm of $\mu_k^+$ grows and its direction stabilizes, moving $\overline{\mathbf{e}}_k^+$ forward along a discriminative direction. By any updates of the gradient of objective $J$, its positive group margin can increase at least $\delta \|\epsilon_+\|$. Therefore, the classification margin of token $\mathbf{x}_k$ can reduce at least $m_i' \geq m_i + \delta \|\epsilon_+\| + \sqrt{2} L_P \gamma$. Therefore, the cross-entropy loss will decrease at least by $\ell_i' \leq \ell_i \cdot \exp\left(-\left(\delta \|\epsilon_+\| + \sqrt{2} L_P \gamma\right)\right)$. This can also be translated into the overall level of optimization acceleration where $\gamma$ can be measured in practice.

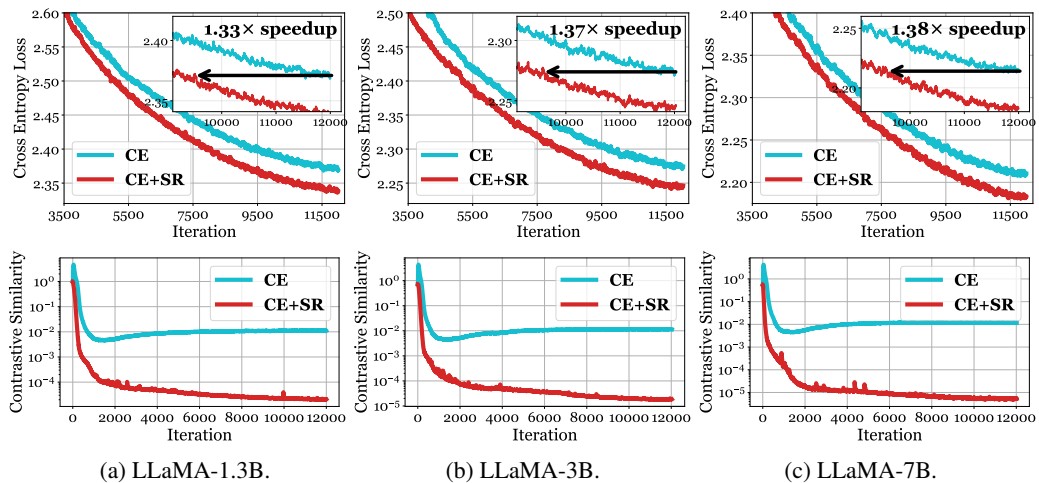

Figure 3: Optimization: cross entropy loss (upper) and similarity regularization loss (lower).

## 5 EXPERIMENTS

In this section, we show the empirical studies of the proposed SIMREG loss. We primarily investigate the advantages in pretraining tasks, including its ability to accelerate the training convergence rate, improve the generalization stability of the downstream tasks, and influence the dynamics of the embedding similarity during training. We also examine its sensitivity to hyperparameters and its behavior. Moreover, we explore the practical effects of inserting the SIMREG loss at different positions within the model. These experiments can provide useful technical guidance for the community.

**Model Backbones.** We mainly select LLaMA (Touvron et al., 2023) and Mixtral (Jiang et al., 2024) as the dense and MoE backbones for pretraining, including the core modules of the mainstream models in the current community, e.g. for RoPE (Su et al., 2024), RMSNorm (Zhang & Sennrich, 2019), and SwiGLU (Shazeer, 2020). We conduct experiments on dense models with 350M, 1.3B, 3B, and 7B parameters, as well as on an MoE model with 8B parameters.

**Training Hyperparameters.** We follow the experimental setups reported in several recent classical LLM pretraining studies (Touvron et al., 2023; Liu et al., 2024a; Jiang et al., 2024; Baidu-ERNIE-Team, 2025) to configure the baseline hyperparameters. We employ the AdamW optimizer (Loshchilov & Hutter, 2019) with $\beta_1 = 0.9$, $\beta_2 = 0.95$, and let the weight decay equals to 0.1. The standard deviation of the weight initialization is set to 0.01. The global batch size is set to 512 for the 350M and MoE-7B models, and 2048 for the 1.3B, 3B, and 7B dense models. The input sequence length is fixed to 2048. For the learning rate schedule, we adopt a 2000-step warm-up phase to linearly increase the learning rate from 0 to $3 \times 10^{-4}$, followed by a cosine decay strategy that gradually reduces it to one-tenth of its peak value. For dense models, we train for 12,500 steps, which corresponds to roughly 13B tokens for the 350M model and 52B tokens for the larger dense models. For MoE models, the total training length is set to 50,000 steps, ensuring exposure to approximately 52B tokens. To avoid the effects caused by loss spikes, we adopt the AdaGC (Wang et al., 2025) to clip gradients for all experiments. For the validation of generalization efficiency, we mainly conduct experiments on the standard downstream benchmarks.

### 5.1 EMPIRICAL STUDIES ON PERFORMANCE

**Higher Convergence.** We first demonstrate the training acceleration of SIMREG in large-scale pretraining tasks. As shown in Figur 3 upper part, on the 1.3B model, the speedup can reach nearly 33%, and after training on 52B tokens, the cross entropy loss can be reduced by about 0.05. On larger-scale models, including the 3B model and the 7B model, SIMREG achieves more than 37% speedup when training reaches 52B tokens, with the final training loss reduced by about 0.03. In the lower part, we present the SIMREG loss. It can be observed that cross-entropy does not impose a mandatory constraint on feature similarity. When training with cross-entropy alone, the feature

Table 1: Generalization efficiency: Zero-shot evaluations on the general downstream tasks.

| | Arc-E | Arc-C | BoolQ | HellaS. | Obqa | Piqa | Mmlu | WinoG. | Sciq | Avg. |
|---|---|---|---|---|---|---|---|---|---|---|
| **LLAMA-350M** | 38.64 | 22.95 | 57.09 | 36.51 | 28.40 | 66.49 | 22.95 | 51.30 | 63.20 | 43.06 |
| ∘ SIMREG | 40.15 | 24.49 | 57.55 | 37.64 | 29.40 | 68.26 | 22.92 | 52.07 | 64.40 | 44.10 |
| ∘ SIMREG-CHUNK | 39.77 | 24.23 | 58.14 | 37.25 | 29.40 | 67.59 | 23.02 | 51.84 | 64.40 | 43.96 |
| **LLAMA-1.3B** | 46.21 | 25.09 | 58.01 | 49.60 | 31.80 | 72.14 | 23.07 | 52.80 | 68.90 | 47.51 |
| ∘ SIMREG | 46.51 | 26.79 | 61.01 | 52.51 | 30.40 | 72.91 | 24.06 | 54.14 | 69.50 | 48.65 |
| ∘ SIMREG-CHUNK | 46.80 | 26.11 | 59.17 | 51.94 | 31.80 | 72.25 | 23.12 | 54.78 | 69.00 | 48.33 |
| **LLAMA-3B** | 48.91 | 27.30 | 58.29 | 55.67 | 33.00 | 74.16 | 23.65 | 55.49 | 73.50 | 50.00 |
| ∘ SIMREG | 50.59 | 28.07 | 58.65 | 57.65 | 33.40 | 74.32 | 23.95 | 56.67 | 75.30 | 50.96 |
| ∘ SIMREG-CHUNK | 50.80 | 27.39 | 62.48 | 58.49 | 33.60 | 73.88 | 22.95 | 55.64 | 73.20 | 50.94 |
| **LLAMA-7B** | 53.07 | 28.84 | 54.07 | 60.41 | 33.80 | 76.12 | 23.79 | 57.30 | 75.70 | 51.45 |
| ∘ SIMREG | 52.57 | 29.01 | 59.79 | 62.01 | 35.80 | 75.14 | 24.47 | 59.04 | 76.20 | 52.67 |
| ∘ SIMREG-CHUNK | 51.60 | 29.69 | 62.39 | 61.80 | 35.80 | 75.46 | 23.51 | 58.72 | 76.00 | 52.77 |
| **MIXTRAL-8×1B** | 48.86 | 29.18 | 54.62 | 59.57 | 34.00 | 73.88 | 24.17 | 56.99 | 72.40 | 50.41 |
| ∘ SIMREG | 51.81 | 28.75 | 60.03 | 62.53 | 35.00 | 75.08 | 23.59 | 54.30 | 74.10 | 51.69 |
| ∘ SIMREG-CHUNK | 52.04 | 28.98 | 60.26 | 62.76 | 35.23 | 73.88 | 23.82 | 54.53 | 73.10 | 51.62 |

similarity undergoes a rapid decline in the early stage, and then gradually tends to stabilize. At this point, the network no longer additionally learns to accelerate classification training by enhancing feature separability. An interesting phenomenon we observe is that, when trained solely with cross-entropy, the similarity regularization value for almost all networks eventually converges to around 0.01, which implies that the average angle between words of different classes is approximately 61.3 degrees. After introducing the SIMREG loss, the embedding similarity decreases significantly, with the regularization loss converging to about 0.00001, indicating that the average angle achieves approximately 74 degrees, which introduces stronger embedding separability.

**Higher Generalization.** Table 1 reports the zero-shot evaluation results across nine general downstream datasets. Overall, introducing the SIMREG loss consistently improves the average performance across different model scales. SIMREG can bring +1.14% average improvement on LLaMA-1.3B, +0.96% on LLaMA-3B, +1.22% on LLaMA-7B, and +1.28% on Mixtral-8×1B. These results highlight that SIMREG provides stable and non-trivial gains as the model scale increases. Moreover, SIMREG achieves the largest single-task gain of +5.72% on BoolQ with LLaMA-7B. Besides BoolQ, we also observe clear improvements on HellaSwag, WinoGrande, and SciQ across multiple scales, showing that it is particularly effective for reasoning-heavy and multi-choice tasks. Importantly, in most cases it achieves the best accuracy among all settings, and the improvements hold consistently across small, medium, and large models.

**Chunk-wise SIMREG.** We primarily investigate the affect of different chunk size in the pretraining in Table 2. We adopt different chunk sizes on the LLaMA-7B pretraining. SIMREG does not require enlarging the chunk size to remain effective, as the incor-

Table 2: Performance of different chunk size.

| | 128 | 256 | 512 | 1024 | 2048 |
|---|---|---|---|---|---|
| Loss | 2.194 | 2.189 | 2.186 | **2.184** | 2.186 |
| Avg. Acc. | 51.46 | 52.52 | 52.54 | **52.77** | 52.67 |

poration of self-similarity provides strong positive signals, yielding the sufficient optimization. This allows SIMREG to naturally support SP training and longer sequence lengths. Our experiments show that a wide range of chunk size (from 256 to 2048) is sufficient to achieve better performance.

## 5.2 HYPERPARAMETER SENSITIVITY

We first grid search $(\tau, \lambda)$ on the 350M model to identify a valid range, followed by a fine-grained search to determine their optimal combinations. Subsequently, we conduct scaling experiments on the 1.3B and 7B models to examine how the optimal choices vary as the model size increases and the corresponding token embedding dimension grows. As shown in Figure 4 (a), to explore the stable results, we grid search the temperature coefficient $\tau$ from $[0.001, 0.003, 0.01, 0.03, 0.1]$ with a $3\times$ skip, and coarsely choose the coefficient $\lambda$ from $[0.01, 0.1, 1, 10, 100]$ with a $10\times$ skip. The valid

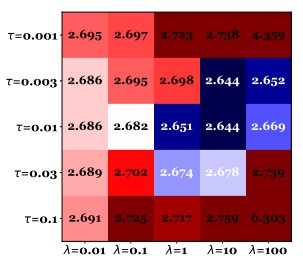
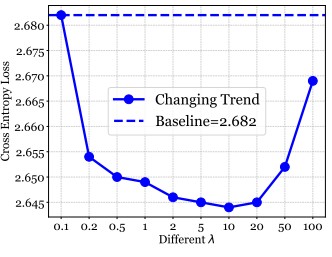
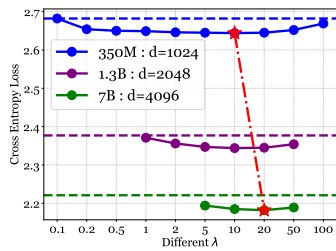

(a) Grid Search of $(\tau, \lambda)$.  (b) Fine-grained Search on $\lambda$.  (c) Scaling with Model Size.

Figure 4: (a) Grid search over hyperparameters $\tau$ and $\lambda$. The blue blocks indicate the values where the final training loss under the corresponding combination $(\tau, \lambda)$ is lower than baseline, with darker colors representing lower losses. (b) We further conduct a fine-grained search over different $\lambda$ values at the generally optimal $\tau = 0.01$, using an approximate $2\times$ scaling ratio. (c) We explore the trends on different $\lambda$ across different model sizes (the red line indicates the optimal trend).

range for $\tau$ is relatively limited, with $0.01$ proving to be a robust selection for all models. $\lambda$ spans a broad effective range from $0.1$ to $100$. Figure 4 (b) presents a fine-grained exploration of $\lambda$, varying it from $0.1$ to $100$ with roughly $2\times$ resolution. The results reveal a stable region between $2$ and $20$. In Figure 4 (c), we explore the scaling of hyperparameters and infer from results across different model sizes how to select optimal hyperparameters. The underlying reason is that as the embedding dimension grows with larger models, the expressive power of the similarity loss decreases, leading to shifts in the optimal hyperparameter choice. Specifically, when the embedding dimension increases, each token is represented in a higher-dimensional space. Therefore, it becomes necessary to increase $\lambda$ to maintain training efficiency. Our experiments confirm this trend, and current results suggest that every time the embedding dimension doubles, the optimal hyperparameter increases by approximately a factor of $\sqrt{2}$ and the optimal $\tau$ can be fixed as $0.01$. We also provide a theoretical analysis to illustrate the reason for this scaling law in Appendix B.2.

## 5.3 OPTIMAL POSITION

In this part, we empirically investigate at which positions in the model applying supervision to embeddings yields the optimal results. We divide the network structure according to its natural layer-wise organization. As shown in Figure 5, it is shown that the performance at intermediate layers is almost negligible. This is expected because tokens in the middle of the network are not simply representations of independent word meanings, but usually carry blended semantic information that integrates

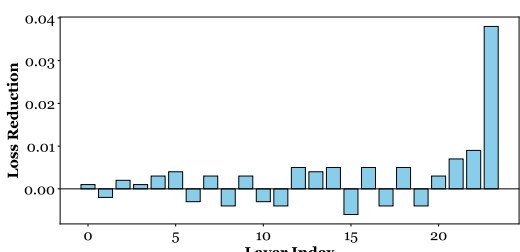

Figure 5: Adopting SIMREG at different layers.

signals from preceding tokens. However, in the ending layers, the model gradually projects these broad token representations into more distinct semantic spaces, and adding similarity regularization at these positions proves to be much more effective. Our experiments further show that applying SIMREG only at the last layer is sufficient to achieve efficient pretraining with almost no overhead.

## 6 CONCLUSION

In this work, we introduced SIMREG, a similarity regularization loss for large-scale pretraining. We investigate that cross-entropy alone fails to enforce embedding consistency, while SIMREG can strengthen representation learning by aligning same-class tokens and separating different classes. Experiments on dense and MoE models demonstrate that SIMREG consistently accelerates convergence by over 30% and improves downstream performance by over 1%. Moreover, it remains robust across model scales and hyperparameter settings. These findings highlight consistency regularization as a promising direction for advancing the efficiency and generalization of LLM pretraining.

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

## A  APPENDIX: STATEMENTS

**The Use of LLMs:** During the preparation of this manuscript, large language models (primarily GPT-5) were used solely to improve the clarity and quality of the text, including correcting grammatical or typographical errors and rephrasing sentences for better readability and fluency. Importantly, they were not involved in any substantive work related to the research contributions of this submission.

**Reproducibility:** We provide a complete experimental pipeline along with detailed hyperparameter choices to ensure reproducibility. All relevant details are thoroughly documented in this submission.

**Ethics Concerns:** This work does not involve any sensitive personal data, human subjects, or animal experiments. The research focuses purely on computational modeling and large-scale pretraining techniques. All datasets used in this study are publicly available and widely adopted in the research community. We have carefully ensured that our methods and findings respect ethical standards and do not pose foreseeable risks of misuse.

## B  APPENDIX: EXPERIMENTS

### B.1  EXPERIMENTAL SETUPS

Here we present the detailed experimental setups in this paper to ensure reproducibility.

**Model Hyperparameters.** We mainly select LLaMA2 (Touvron et al., 2023) and Mixtral (Jiang et al., 2024) as the dense and MoE backbones for pretraining, including the core modules of the mainstream models in the current community, e.g. for RoPE (Su et al., 2024), RMSNorm (Zhang & Sennrich, 2019), and SwiGLU (Shazeer, 2020). We follow the common practices in the community to scale models of different sizes, and the detailed configurations are shown in Table 3.

Table 3: Model Hyperparameters.

|  | Experts | Layers | Attention heads | Embedding dim | FFN hidden size |
|---|---|---|---|---|---|
| LLaMA2-350M | 1 | 24 | 16 | 1024 | 2371 |
| LLaMA2-1.3B | 1 | 24 | 32 | 2048 | 5461 |
| LLaMA2-3B | 1 | 26 | 32 | 3072 | 8640 |
| LLaMA2-7B | 1 | 32 | 32 | 4096 | 11008 |
| Mixtral-8×1B | 8 | 24 | 32 | 2048 | 5632 |

**Training Hyperparameters.** We follow the experimental setups reported in several recent classical LLM pretraining studies (Touvron et al., 2023; Liu et al., 2024a; Jiang et al., 2024) to configure the baseline hyperparameters, ensuring comparability with prior work. Specifically, we employ the AdamW optimizer (Loshchilov & Hutter, 2019) with $\beta_1 = 0.9$ and $\beta_2 = 0.95$, and a weight decay of $0.1$. The standard deviation of weight initialization is set to $0.01$. To balance efficiency and stability, we use a global batch size of $512$ for the 350M and MoE-1×8B models, and $2048$ for the 1.3B, 3B, and 7B dense models, while the input sequence length is fixed at $2048$.

For the learning rate schedule, we adopt a 2000-step warm-up phase that linearly increases the learning rate from 0 to $3e$-4, followed by a cosine decay strategy that gradually reduces it to one-tenth of its peak value. Regarding training length, dense models are trained for $12,500$ steps, corresponding to roughly 13B tokens for the 350M model and 52B tokens for the larger dense models. In contrast, MoE models are trained for $50,000$ steps to ensure comparable exposure of approximately 52B tokens. Finally, to mitigate potential instabilities caused by loss spikes, we adopt AdaGC (Wang et al., 2025) for adaptive gradient clipping. We summarize the details in Table 4.

Table 4: Training Hyperparameters.

|  | batchsize | seqlen | learning rate | $\lambda_{\mathrm{w}}$ | $\beta_1$ | $\beta_2$ | clip-$\lambda$ | clip-$\beta$ |
|---|---|---|---|---|---|---|---|---|
| LLaMA-350M | 512 | 2048 | 4e-4 $\rightarrow$ 4e-5 | 0.1 | 0.9 | 0.95 | 1.04 | 0.99 |
| LLaMA-1.3B | 2048 | 2048 | 3e-4 $\rightarrow$ 3e-5 | 0.1 | 0.9 | 0.95 | 1.04 | 0.99 |
| LLaMA-3B | 2048 | 2048 | 3e-4 $\rightarrow$ 3e-5 | 0.1 | 0.9 | 0.95 | 1.04 | 0.99 |
| LLaMA-7B | 2048 | 2048 | 3e-4 $\rightarrow$ 3e-5 | 0.1 | 0.9 | 0.95 | 1.04 | 0.99 |
| Mixtral-8$\times$1B | 512 | 2048 | 3e-4 $\rightarrow$ 3e-5 | 0.1 | 0.9 | 0.95 | 1.04 | 0.99 |

**Specific Hyperparameters.** Our proposed loss function is primarily characterized by two key hyperparameters, the temperature $\tau$ and the coefficient $\lambda$. We conduct extensive grid search experiments ($\tau \in [0.001, 0.01, 0.1]$ and $\lambda \in [0.2, 0.5, 1, 2, 5, 10, 20, 50, 100]$) on the 350M model to determine the effective range of these hyperparameters, and validate them on larger models according to scaling theory. The simple settings of $\tau = 0.01$ and $\lambda = 10$ are sufficient to achieve good performance for most experiments. To better adapt to the model scaling, we explore a more refined yet simple strategy to determine the selections, which is detailed in Sec.B.2.

**Evaluations.** To ensure a fair comparison, we conduct all evaluations on EleutherAI/lm-evaluation benchmark (Gao et al., 2024). We mainly evaluate the performance of the pretrained model on downstream tasks of arc_easy, arc_challenge, openbookqa, boolq, hellaswag, piqa, winogrande, mmlu, sciq (general reasoning ability) and the domain-specific downstream tasks of gsm8k, drop, race, squadv2, nq_open, humaneval, mbpp (three domains: math, code, and reading comprehension).

**Training Resources.** We conduct experiments on H800 GPUs. Pretraining the 350M model on 13B tokens requires approximately 56 GPU hours per experiment and the 7B model on 52B tokens takes over 2,000 GPU hours per experiment.

## B.2 HOW TO SCALE HYPERPARAMETERS ON LARGE MODELS?

In this part, we introduce a refined hyperparameter tuning mechanism to accommodate model scaling. Before introducing it, we first demonstrate the relationship between the representation ability of our SIMREG loss and the dimensionality of embeddings in the model. The SIMREG loss regularizes pretraining by leveraging the token embedding similarity between pairs of tokens. By assuming $\mathbf{x}, \mathbf{y} \in \mathbb{R}^d$ are independent and identically distributed as isotropic random variables, e.g., $\mathbf{x}, \mathbf{y} \sim \mathcal{N}(0, I_d)$. Thus, we consider their cosine similarity $z = \frac{\langle \mathbf{x}, \mathbf{y} \rangle}{\|\mathbf{x}\| \cdot \|\mathbf{y}\|} \in [-1, 1]$. Without loss of generality, we can assume $\frac{\mathbf{y}}{\|\mathbf{y}\|} = (1, 0, \cdots, 0)$ as the first basis of the spherical space $S^{d-1}$. Then the distribution of $z$ can be transferred to the study of the first coordinate of $v \sim \mathrm{Uinf}\left(S^{d-1}\right)$. Substitute $v$ into the iterative form of spherical coordinates $v = (\cos\theta, \sin\theta \cdot \zeta)$ where $\zeta \in S^{d-2}$. According to the decomposition of the spherical surface unit, we have $d\sigma_{d-1}(v) = \sin^{d-2}(\theta)\, d\theta\, d\sigma_{d-1}(\zeta)$ and the marginal density of the polar angle:

$$f_p(\theta) = \frac{1}{|S^{d-1}|} \int_{S^{d-2}} \sin^{d-2}(\theta)\, d\sigma_{d-2}(v) = \frac{|S^{d-2}|}{|S^{d-1}|} \sin^{d-2}(\theta).$$

Then we consider the variable $z$. Due to the first coordinate $z = v_0 = \cos(\theta)$, we have:

$$f_p(z) = f_p(\theta) \left| \frac{d\theta}{dz} \right| = \frac{|S^{d-2}|}{|S^{d-1}|} \cdot \frac{\sin^{d-2}(\theta)}{\sin(\theta)} = \frac{|S^{d-2}|}{|S^{d-1}|} \left(1 - z^2\right)^{\frac{d-3}{2}} = \frac{\Gamma(\frac{d}{2})}{\sqrt{\pi}\Gamma(\frac{d-1}{2})} \left(1 - z^2\right)^{\frac{d-3}{2}}.$$

It is easy to check that $\mathbb{E}[z] = 0$ and $\mathbb{E}[z^2] = \frac{1}{d}$. Therefore, as the model size increases and the embedding dimensionality changes from $d_0$ to $d_1$, the capacity of SIMREG loss decreases by a factor of $\sqrt{\frac{d_1}{d_0}}$. To preserve the representation capability, we can revise the $\lambda$ coefficient.

We next investigate the feasibility of this scaling method from an empirical perspective. We separately sweep the hyperparameters and report the evaluation perplexity (ppl) at the end of training.

Table 5: Validation perplexity (generalization performance) of different $(\tau, \lambda_{\text{reg}})$.

|  | $\tau = 0.005$ | $\tau = 0.01$ | $\tau = 0.02$ | $\tau = 0.05$ | $\tau = 0.1$ |
|---|---|---|---|---|---|
| $\lambda_{\text{reg}} = 0$ (baseline) | | | 15.06 | | |
| $\lambda_{\text{reg}} = 0.1$ | 14.98 | 15.01 | 14.92 | 14.95 | 14.98 |
| $\lambda_{\text{reg}} = 0.2$ | 14.50 | 14.38 | 14.36 | 14.79 | 14.81 |
| $\lambda_{\text{reg}} = 0.5$ | 14.45 | 14.41 | 14.43 | 14.54 | 14.88 |
| $\lambda_{\text{reg}} = 1$ | 14.41 | 14.45 | 14.46 | 14.67 | 14.71 |
| $\lambda_{\text{reg}} = 2$ | 14.39 | 14.41 | 14.49 | 14.65 | 14.74 |
| $\lambda_{\text{reg}} = 5$ | 14.36 | 14.36 | 14.37 | 14.35 | 15.13 |
| $\lambda_{\text{reg}} = 10$ | 14.34 | **14.25** | 14.36 | 14.44 | 15.65 |
| $\lambda_{\text{reg}} = 20$ | 14.41 | 14.29 | 14.44 | 14.59 | - |
| $\lambda_{\text{reg}} = 50$ | 14.74 | 14.53 | 14.55 | 14.78 | - |

The optimal range and variation trend in Table 5 and Figure 4 are almost identical to those observed in the optimization process, **indicating that the improvements brought by the SIMREG loss in both optimization and generalization are consistent**. The optimal choice of $\tau$ remains concentrated around $0.01$. Next, we evaluate models of different scales (primarily with increased embedding hidden sizes), while keeping $\lambda$ fixed at $0.01$.

Table 6: Optimal validation perplexity (generalization performance) of different model size.

|  | 350M ($d = 1024$) | 1.3B ($d = 2048$) | 3B ($d = 3072$) | 7B ($d = 4096$) |
|---|---|---|---|---|
| $\lambda_{\text{reg}} = 0$ (baseline) | 15.06 | 10.72 | 9.70 | 8.99 |
| $\lambda_{\text{reg}} = 5$ | 14.36 | 10.46 | 9.50 | 8.92 |
| $\lambda_{\text{reg}} = 10$ | **14.25** | **10.41** | 9.46 | 8.84 |
| $\lambda_{\text{reg}} = 20$ | 14.29 | 10.42 | **9.44** | **8.78** |
| $\lambda_{\text{reg}} = 50$ | 14.33 | 10.49 | 9.49 | 8.81 |

It can be observed that the trend largely aligns with our hypothesis. Therefore, we propose the following estimation method for the optimal hyperparameters:

$$\tau = 0.01, \ \lambda_{\text{reg}} \approx 10 \times \sqrt{\frac{d}{1024}},$$

where $d$ is the dimension of the hidden-size of the token embedding. Of course, the scale of the model also affects the results. In practice, a simple grid search within this range of choices can be performed to identify the optimal combination.

## B.3 SIMREG LOSS CURVES

In this section, we mainly present the variations of the SIMREG loss. We explore the limitations of cross-entropy in LLM pretraining, namely, that it cannot achieve better classification performance simply by further reducing feature separability. This is because cross-entropy focuses solely on aligning predictions with ground-truth labels, while leaving the underlying structure of token embeddings insufficiently constrained. As the model scales up, this weakness becomes more pronounced: embeddings of the same class may still scatter in the representation space, leading to instability in optimization and slower convergence. By contrast, the SIMREG loss explicitly regularizes intra-class consistency and inter-class separation, complementing cross-entropy with a more direct control of embedding geometry. This additional constraint not only improves convergence speed but also yields more robust generalization in downstream tasks.

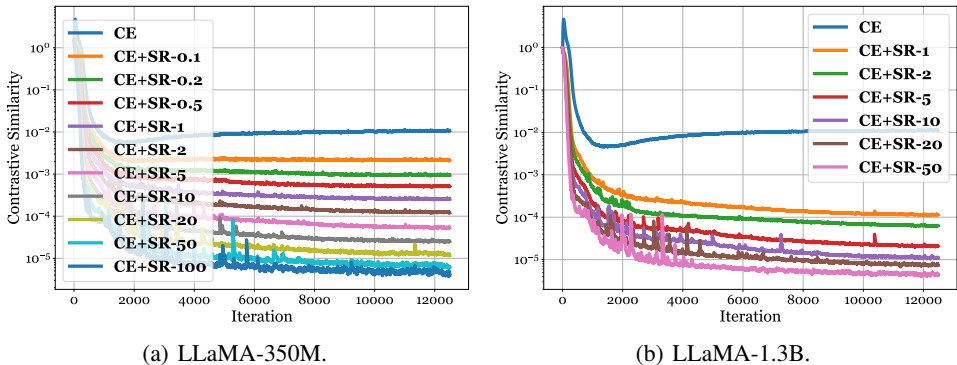

(a) LLaMA-350M.  (b) LLaMA-1.3B.

Figure 6: The training curve of the SIMREG loss.

Figure 6 illustrates the loss behavior when increasing the coefficient of the SIMREG loss. It can be observed that even with a large weighting ratio, SIMREG does not cause the training to diverge. At the same time, we also note that the feature consistency loss exhibits a strictly monotonic trend. This phenomenon suggests that SIMREG serves as a stable regularization term: rather than interfering with the optimization of cross-entropy, it progressively strengthens the alignment of token embeddings as its weight grows. In practice, this means that a wide range of coefficient values can be applied without destabilizing training, making SIMREG highly robust and easy to integrate into large-scale pretraining pipelines.

**Trade-off of $\lambda$.** Although we generally hope that greater feature separability will lead to better performance, the pretraining process involves not only learning representations but also learning classification. If $\lambda$ is increased without bound, the weight of SIMREG may eventually become too dominant and interfere with the optimization of cross-entropy. This phenomenon can be directly observed from the changes in gradient behavior, which provide an intuitive reflection of the trade-off between the two objectives. Table 7 shows the comparison clearly illustrates the effects of cross-entropy and SIMREG under different parameter settings.

Table 7: Changing trend of CrossEntropy and SIMREG loss on different $\lambda$.

|  | $\lambda = 1$ | $\lambda = 2$ | $\lambda = 5$ | $\lambda = 10$ | $\lambda = 20$ | $\lambda = 50$ |
|---|---|---|---|---|---|---|
| CrossEntropy | 2.38 | 2.35 | 2.32 | **2.30** | 2.33 | 2.35 |
| SIMREG ($\times$1e-6) | 104.6 | 56.2 | 18.3 | 9.5 | 6.0 | 3.1 |

### B.4 VISUALIZATION OF THE TOKEN EMBEDDING SIMILARITY

Here we provide more visualization demos of the true pretraining data samples from C4 dataset on LLaMA-7B pretraining by 52B tokens.

**Text1:** [ so I'm not sure if there's anything holding the back. I do not think there is by wiggling on it but could possibly have a strap or the like. I would think there must be a way to remove the panel blocking the bottom of the washer. We installed our own washer and used the clips mentioned in the previous post. Here is a PDF file on how they are used and what they look like. You may want to run your fingers over the entire carpeted lip ... typically, the ]

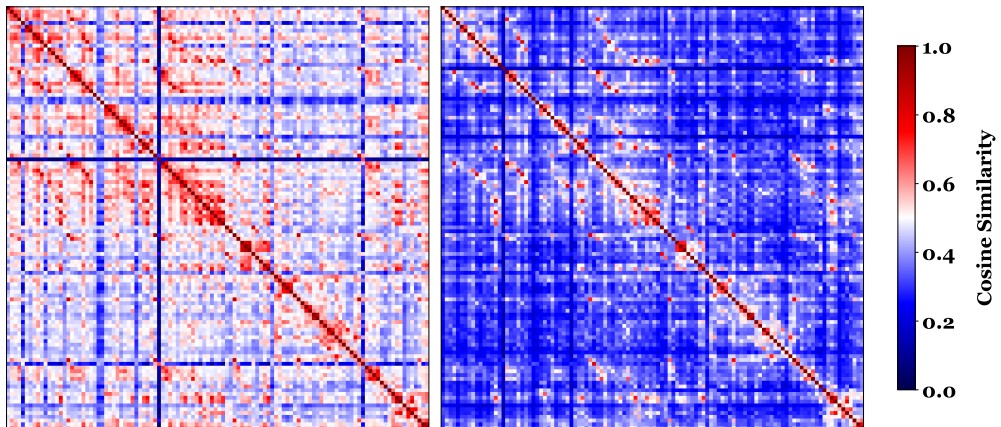

Figure 7: The averaged cosine similarity values are 0.488 (CrossEntropy only - left) and 0.354 (CrossEntropy + SIMREG - right).

**Text2:** [ manufacturer runs screws into the floors/cabinets and the heads are buried in the carpet. There are two screws with square heads in the top of the carpet. Have you tired to do the recommended procedure to clean the lint out of the drain. 1. Run the unit without clothes and with the dry time off on cycle # 11. 2. When the water stops entering the unit push and hold the start button until all the lights come on then release the button. ]

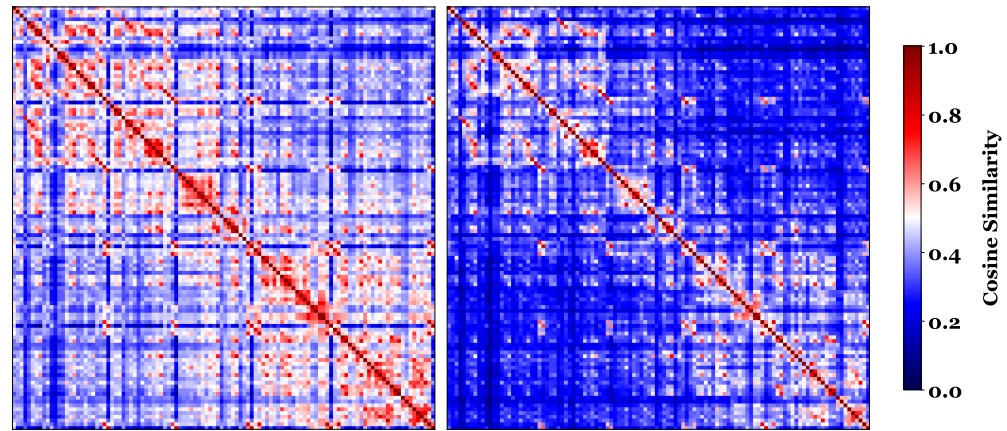

Figure 8: The averaged cosine similarity values are 0.445 (CrossEntropy only - left) and 0.333 (CrossEntropy + SIMREG - right).

## C APPENDIX: THEORETICAL ANALYSIS

In this section, we mainly demonstrate the theoretical understanding to show how the SIMREG loss improves the convergence and generalization efficiency. To this end, we first establish the fundamental properties of the proposed objective and analyze its impact on representation learning. We then present rigorous bounds and intuitive explanations that highlight its advantages over conventional cross-entropy training. These insights not only provide a deeper understanding of why SIMREG is effective but also offer useful guidance for its broader application in large-scale pretraining.

### C.1 RELATIONSHIP BETWEEN EMPIRICAL LOSS AND MARGINS

We first introduce the simplified modeling and corresponding notations of the LLM pretraining. Without loss of generality, we decompose the model into two simple parts. The first part is the front-end structure, which takes the raw data as input and outputs the embedding representations. The second part is the back-end structure, which transforms the embeddings into logits, followed by a cross-entropy loss function. We denote $X = [x_1, x_2, \cdots, x_n] \in \mathbb{R}^{n \times d}$ as the embeddings and $Z = f_h(X) \in \mathbb{R}^{n \times c}$ as the logits. The category label are denoted by $Y = [y_1, y_2, \cdots, y_n] \in \{1, 2, \cdots, C\}^n$. For the sample-wise cross entropy loss, we have:

$$\ell(x_i, y_i) = -z_{i,y_i} + \log\left(\sum_{j=1}^{K} e^{z_{i,j}}\right).$$

The empirical loss is $L = \frac{1}{n}\sum_{i=1}^{n} \ell(x_i, y_i)$. Then we consider the margin value in multi-class classification, which is also the joint gaps of different categories $m_i = z_{i,y_i} - \max_{j \neq y_i} z_{i,j}$. Therefore, we have:

$$\ell(x_i, y_i) = \log\left(1 + \sum_{j \neq y_i} e^{-(z_{i,y_i} - z_{i,j})}\right) \leq \log\left(1 + (C-1)e^{-m_i}\right) \leq (C-1)e^{-m_i},$$

where the empirical loss is $L = \frac{1}{n}\sum_{i=1}^{n} \ell(x_i, y_i) \leq \frac{C-1}{n}\sum_{i=1}^{n} e^{-m_i}$. Generally, if the classification margins of all samples are increased by at least $\Delta \geq 0$, the loss will be multiplicatively reduced by a factor of $e^{-\Delta}$.

### C.2 EQUIVALENT CONSTRAINT OF THE SIMREG LOSS

Here we learn how the SIMREG loss affect the embeddings and the model performance. Here we let each embedding $\mathbf{e}_i = r_i \mathbf{a}_i$ where $r_i = \|\mathbf{e}_i\| \geq 0$ is the magnitude and $\mathbf{a}_i$ is the normalized embedding. SIMREG loss evaluates the exponential of the cosine similarity of two embeddings. Its core focus lies in the geometric information of the term $a$. To learn the performance of the SIMREG, for each label $y_i$, we define a positive set $\mathcal{P}_i = \{a_j : y_j = y_i\}$ and a negative set $\mathcal{N}_i = \{a_j : y_j \neq y_i\}$. The union of $\mathcal{P}_i$ and $\mathcal{N}_i$ always combines a complete sequence.

To understand the performance of SIMREG in detail, we first introduce a general kernal function $\kappa(\mathbf{u}, \mathbf{v}) = \exp(\mathbf{u}^\top \mathbf{v})$, which admits the Maclaurin series $\kappa(\mathbf{u}, \mathbf{v}) = \sum_{m=0}^{\infty} \frac{(\mathbf{u}^\top \mathbf{v})^m}{m!}$. It is a positive definite kernel on the unit sphere. By introducing an explicit map: $h : \mathbb{S}^{d-1} \to \mathcal{H}$ on the symmetric tensor powers:

$$h(\mathbf{u}) = \left[1, \frac{1}{\sqrt{\pi}}\mathbf{u}, \frac{1}{\sqrt{2!\pi^2}}\text{vec}\left(\mathbf{u}^{\otimes 2}\right), \frac{1}{\sqrt{3!\pi^3}}\text{vec}\left(\mathbf{u}^{\otimes 3}\right), \cdots\right], \tag{9}$$

thus we have the transformation of $\langle h(\mathbf{u}), h(\mathbf{v})\rangle = \kappa(\mathbf{u}, \mathbf{v})$. The mapping $h$ is to construct a linear expansion of $\kappa$ in the reproducing kernel Hilbert space (RKHS) $\mathcal{H}$. Therefore, we have:

$$\log\left(\sum_{i \in \mathcal{P}_k} \exp\left(\mathbf{e}_k^\top \mathbf{e}_i\right)\right) = \log\left(\sum_{i \in \mathcal{P}_k} \langle h(\mathbf{e}_k), h(\mathbf{e}_i)\rangle\right) = \log\left(\langle h(\mathbf{e}_k), \mu_k^+\rangle\right) + \log\left(|\mathcal{P}_k|\right),$$

where $\mu_k^+ = \frac{1}{|\mathcal{P}_k|}\sum_{i \in \mathcal{P}_k} h(\mathbf{e}_i)$ is the positive kernel means. Here $|\mathcal{P}_k|$ can be considered as a offset to scale the positive samples. The theoretical analysis can be symmetrically extended to negative samples, yielding an equivalent conclusion.

Therefore, the SIMREG loss consider the difference between teh positive and negative set by:

$$\min_{\mathbf{e}=f_E(\mathbf{x})} \quad J = \mathbb{E}_{\mathbf{x}} \log \left( \frac{\langle h(\mathbf{e}_k), \mu_k^- \rangle}{\langle h(\mathbf{e}_k), \mu_k^+ \rangle} \right) + \log \left( \frac{|\mathcal{N}_k|}{|\mathcal{P}_k|} \right).$$

The ratio of positive to negative samples only affects the scale of the loss, but does not alter the primary optimization objective of the first term. It pushes the anchor direction to align with the positive kernel mean and to anti-align with the negative kernel mean. It also nudges the group means themselves: positives move toward anchors that they are already close to, and negatives move away in the RKHS sense. We also have the nearest positive prototype for each class:

$$\max_{\|\mathbf{e}\|} \langle h(\mathbf{e}), \mu_k^+ \rangle = \|h(\mathbf{e})\| \|\mu_k^+\| = \kappa(\mathbf{e}, \mathbf{e}) \|\mu_k^+\| = \sqrt{e} \|\mu_k^+\|.$$

The same, the $\sqrt{e}$ scaling also hold for the negative set. Beyond the optimization objective itself, we can further consider the problem from the perspective of gradient directions to refine the learning target. By considering the Fréchet gradient, we have:

$$\nabla_{h(\mathbf{e}_k)} J = \frac{\mu_k^-}{\langle h(\mathbf{e}_k), \mu_k^- \rangle} - \frac{\mu_k^+}{\langle h(\mathbf{e}_k), \mu_k^+ \rangle}.$$

Generally, $\mu_k^- \neq \mu_k^+$. From the gradient expression, we can see that the optimization dynamics naturally combine both "attractive" and "repulsive" effects. Specifically, the first term pushes the representation $h(\mathbf{e}_k)$ away from the negative center $\mu_k^-$, while the second term pulls it closer to the positive center $\mu_k^+$. As a result, the overall update direction is shaped by the joint effect of being attracted to positives and repelled from negatives, thereby optimizing the representation space effectively. From the above two perspectives, it is clear that SIMREG enforces feature consistency alignment in the RKHS sense.

## C.3 CENTER-ALIGNED EMBEDDINGS CAN ENHANCE OPTIMIZATION

Then we consider the performance of the center-aligned embedding. To learn the transferred impact from the mapping $h(\mathbf{e}_k)$ to vanilla variable $\mathbf{e}_k$, we first consider the normalized $\mathbf{a}_k$ term, where the cosine similarity can be considered as $\mathbf{a}_k^\top \mathbf{a}_j$. To simplify the notation, we additionally define the weighted average direction of a variable $\mathbf{a}$ over its associated positive and negative sets by $\mathbf{v}_k^+ = \frac{1}{\|\mathcal{P}_k\|} \sum_{i \in \mathcal{P}_k} \exp(\mathbf{a}_k^\top \mathbf{a}_i) \mathbf{a}_i$ and $\mathbf{v}_k^- = \frac{1}{\|\mathcal{N}_k\|} \sum_{j \in \mathcal{N}_k} \exp(\mathbf{a}_k^\top \mathbf{a}_j) \mathbf{a}_j$. Similarly, we also define the loss of positive set and negative set as $P_k$ and $N_k$. Therefore, we have the following gradient form:

$$\nabla_{\mathbf{a}_k} L_{\mathrm{sr}} = \frac{N_k}{P_k + N_k} \left( \mathbf{v}_k^- - \mathbf{v}_k^+ \right).$$

Since the $\mathbf{a}_k$ is constrainted by $\|\mathbf{a}_k\| = 1$, the true update direction is obtained by projecting the gradient onto the tangent space: $-\prod_{\mathbf{v}_k} \nabla_{\mathbf{a}_k} L_{\mathrm{sr}} = -\frac{N_k}{P_k + N_k} \left( I - \mathbf{a}_k \mathbf{a}_k^\top \right) \left( \mathbf{v}_k^- - \mathbf{v}_k^+ \right)$. Next, we analyze how the gradient dynamics associated with the positive sample set vary along the update direction. This dynamic essentially characterizes how strongly the representation is pulled toward the positive center during optimization. A larger value indicates that the update direction aligns well with the attraction force from positive samples, thereby accelerating convergence. Conversely, a smaller value reflects weaker alignment, suggesting limited contribution from positive samples in shaping the optimization trajectory. For the positive sample loss, we obtain (for clarity of exposition, we omit constant scalar terms):

$$\frac{d}{dt} \|\mathbf{a}_k - \mathbf{v}_k^+\|^2 = 2 \left( \mathbf{a}_k - \mathbf{v}_k^+ \right)^\top \left( I - \mathbf{a}_k \mathbf{a}_k^\top \right) \mathbf{v}_k^+ - \underbrace{2 \left( \mathbf{a}_k - \mathbf{v}_k^+ \right)^\top \left( I - \mathbf{a}_k \mathbf{a}_k^\top \right) \mathbf{v}_k^-}_{\text{negative perturbation}}.$$

When treating the update on the negative sample set as a small perturbation to that on the positive samples, we have $\frac{d}{dt} \|\mathbf{a}_k - \mathbf{v}_k^+\|^2 \leq 2 \left( \mathbf{a}_k^\top \mathbf{v}_k^+ \right)^2 - \|\mathbf{v}_k^+\|^2 \leq 0$. Similarly, the gradient dynamics on the negative sample set can be obtained as $\frac{d}{dt} \|\mathbf{a}_k - \mathbf{v}_k^-\|^2 \geq 0$. In conclusion, taking a small step along the tangent update direction inherently drives the representation closer to the weighted center of the positive class while simultaneously pushing it away from that of the negative class. In other

words, such updates reinforce the consistency among positive samples and reduce the influence of negatives, thereby shaping a clearer separation in the feature space. Importantly, this property does not rely on any assumptions about the underlying functional form, but rather arises directly from the optimization objective itself, ensuring both generality and robustness. To further refine the update dynamics, a temperature coefficient can be introduced as a scaling factor. By adjusting the sharpness of the similarity distribution, the temperature effectively controls the relative strength of attraction toward positive samples and repulsion from negative samples. In particular, incorporating a temperature into the formulation normalizes the gradient magnitudes and ensures that the update direction satisfies the desired balance condition between positive and negative contributions. This modification not only stabilizes training but also enhances the flexibility of the loss function in adapting to different representation scales. This result can be directly extended from the normalized variables to the original embedding variables $\mathbf{e}$, thereby completing the proofs.

