# OpenReview forum: "SimReg: Achieving Higher Convergence and Generalization in the LLM Pretraining via Embedding Similarity Regularization"
_ICLR.cc/2026/Conference — Submitted to ICLR 2026_

### Official Review · Reviewer_ueGi · 2025-10-29

**Soundness:** 2
**Presentation:** 3
**Contribution:** 2
**Rating:** 2
**Confidence:** 4

**Summary:**

The paper introduces SimReg, a contrastive regularization loss that addresses a key limitation of cross-entropy training in LLMs: embeddings predicting the same token can vary substantially due to different contexts, and cross-entropy stops enforcing embedding separability once basic classification is achieved. SimReg explicitly pulls together embeddings with the same prediction target while pushing apart those with different targets, which theoretically enlarges classification margins and can accelerate optimization. The method includes practical design choices for stability and scalability, with hyperparameters that scale predictably with model size. Experiments show that SimReg achieves faster convergence and higher model quality.

**Strengths:**

- The paper addresses sample efficiency in LLM pre-training pipelines, a key factor for developing more useful models and reducing operating costs.
- The idea of using cosine-similarity of readily available embeddings as an additional component of the loss enriches the optimization process and can yield overall better performance

**Weaknesses:**

- My main concern is that the paper does not look into other loss-regularizing/-reweighting techniques that augment the optimization process [1, 2, 3, 4]. In particular, I would be interested better understanding how the cost dynamics compare.
- Chunking seems to be a large part of making Simreg practically applicable. How should the chunk size have to be configured in context of other hyperparameters like the learning rate?
- I am generally missing the stanard deviation in all plots and tables. Since the performance improvements of Simreg on the various benchmarks in Table 1 appear to be quite small, getting a full picture requires the standard deviation.

**Minor remarks:**
- Line 79: The word "only" seems to be misplaced
- Line 161: Sentence starting with "The model typically employ..." Here is an s missing.
- Line 262: typo "kernal"

**Sources**
[1] Text and Code Embeddings by Contrastive Pre-Training,  Neelakantan et al., 2022
[2] Rho-1: Not All Tokens Are What You Need, Lin et al., 2024
[3] DoGE: Domain Reweighting with Generalization Estimation, Fan et al., 2023
[4] Dynamic Loss-Based Sample Reweighting for Improved Large Language Model Pretraining, Sow et al., 2025

**Questions:**

Please see weaknesses

---

### Official Review · Reviewer_Vb22 · 2025-10-30

**Soundness:** 3
**Presentation:** 3
**Contribution:** 2
**Rating:** 4
**Confidence:** 4

**Summary:**

The authors propose SimReg, a consistency regularisation term designed to enhance the representational capacity of the model. The objective of SimReg is to effectively pull embeddings of the same class together while push them apart from different classes. The authors also provide a theoretical analysis attempting to establish a formal link between the SimReg and the enhancement of classification margins.

**Strengths:**

1. Significance: The work tackles a core challenge in LLM development: improving pretraining efficiency and model generalisation.
2. Theoretical Foundation: The authors provide a valuable theoretical analysis (Lemmas 4.1 and 4.2) that formally connects the SimReg to the enlargement of classification margins, providing a theoretical rationale for SimReg.
3. Empirical Results: The experimental validation demonstrates improvements in both convergence speed (reportedly over 30%) and downstream task performance (over 1% on average in zero shot benchmarks).

**Weaknesses:**

1. Gap in Theoretical Analysis: The abstract claims "Our comprehensive theoretical analysis elucidates how SIMREG improves both classification margins and generalization in the pretraining stage". However, the provided Lemmas and the discussion in Section 4 only establishes a link between the SimReg objective and the margin. The crucial link between an enlarged margin and improved generalisation ability (i.e., test error/performance on downstream tasks) is supported only by empirical results rather than a formal proof, which does not constitute a comprehensive theoretical analysis.
2. Missing Experimental Comparisons: The paper lacks experimental comparisons with existing contrastive or consistency-based methods that could be applied to large language model pretraining.
3. Insufficient Discussion of Results: The paper does not sufficiently analyse or discuss certain anomalous experimental outcomes. For instance, in Table 1 (LLaMA 7B), the chunked approximation (SIMREG CHUNK) marginally outperforms the full SIMREG (52.77 vs 52.67), which is counter intuitive given that it is an approximation. Furthermore, on specific tasks (like Arc E for LLaMA 7B), SIMREG performs worse than the baseline. A deeper investigation into these phenomena would strengthen the paper's conclusions.
4. Lack of Quantitative Overhead Analysis: While a chunking method is proposed to reduce complexity, this claim is not substantiated with a quantitative analysis of the actual impact on training time, memory, or computational resources, particularly in long sequence scenarios.
5. Minor Typos: The paper contains several typos. For example:
(1) In the definition of the function g (around line 288), ${y_{i}}$ should probably be ${y_k}$, $f_P(k)$ should probably be $f_P(\mathbf e_k)$;
(2) In the discussion of the margin (around line 322), the text states the margin can "reduce". This should likely be "increase" or "improve". The index $i$ in $m_i'$ should probably be $k$.

**Questions:**

1. Could the authors elaborate on the theoretical link between the margin illustrated in the paper and generalisation ability (i.e., test error/performance on downstream tasks) to fully support the theoretical contribution claimed in the abstract?
2. Could the authors include additional experimental results and analyses comparing SimReg with other contrastive or consistency-based methods? This would strengthen the support for the method’s empirical performance and help demonstrate that the proposed contrastive learning strategy is particularly well-suited for LLM pre-training.
3. Do the authors have an explanation or hypothesis for the anomalous results observed (e.g., SIMREG CHUNK outperforming SIMREG, or SIMREG underperforming the baseline on specific tasks)?
4. Could the authors provide a quantitative analysis of the computational overhead (e.g., percentage increase in training time or memory usage) introduced by SimReg?

---

### Official Review · Reviewer_JGnN · 2025-11-01

**Soundness:** 3
**Presentation:** 3
**Contribution:** 3
**Rating:** 6
**Confidence:** 3

**Summary:**

The paper introduces SIMREG, a similarity regularization loss that improves convergence and generalization in large-scale language model pretraining. By complementing cross-entropy, SIMREG enforces higher intra-class embedding similarity and inter-class separability through a contrastive cosine-based objective. It integrates efficiently into both dense and Mixture-of-Experts architectures and supports chunk-wise parallelization for long sequences. Theoretical analysis links SIMREG to enlarged classification margins and faster optimization, while experiments on LLaMA and Mixtral models show 30–37% faster convergence and about 1% average improvement in zero-shot downstream benchmarks. Overall, SIMREG provides a simple yet effective framework for enhancing embedding consistency during LLM pretraining.

**Strengths:**

- SIMREG successfully complements CE by imposing mandatory constraints on feature similarity, resulting in significantly enhanced separability.
- SIMREG shows a wide range of applicability regarding its weighting hyperparameter (λ). The paper also provides critical practical insights, recommending that SIMREG is most effective when applied solely at the output of the final layer of the network.
- The inclusion of a detailed theoretical analysis explaining how the SIMREG loss translates into increased classification margins and accelerated loss reduction provides strong justification for the observed empirical performance boosts.

**Weaknesses:**

- While the paper mentions that applying SIMREG only at the last layer results in almost no overhead, the actual runtime or GPU cost increase relative to the baseline training step is not explicitly quantified.
- The derivation of the hyperparameter scaling law for λ relies on treating embeddings x,y as independent and identically distributed isotropic random variables. This assumption may overly simplify the highly non-isotropic and context-dependent nature of token embeddings in real LLMs, potentially limiting the generalizability of the scaling guidance.
- The paper notes that LLM pretraining data exhibits a pronounced long-tail effect where frequent tokens dominate the CE loss. While SIMREG aims to stabilize representations, it is not clearly discussed how the required positive sets are handled for extremely rare tokens.

**Questions:**

- Can the authors provide an analysis or visualization demonstrating how SIMREG specifically influences the clustering and separability of tokens belonging to the heavy long-tail compared to high-frequency tokens?
- Could the authors provide a quantitative comparison of the per-step training time for the CE-only baseline versus the proposed CE+SIMREG configuration?

---

### Meta-Review · Area_Chair_wnDs · 2026-01-02

**Summary:**

Reviewers raised questions regarding the following aspects
1. Lack of quantitative analysis on the overhead.
2. Missing details regarding theoretical discussions and experiments.
3. Not enough comparison with prior works.

**Reviewer Concerns:**

Reviewers didn't provide a rebuttal, so all the concerns are still outstanding.

**Reviewer Scores:**

Reviewers will keep their scores since no rebuttal was provided by the authors.

---

### Decision · Program_Chairs · 2026-01-26

Reject